# DNA Sequence and Structure under the Prism of Group Theory and Algebraic Surfaces

**DOI:** 10.3390/ijms232113290

**Published:** 2022-10-31

**Authors:** Michel Planat, Marcelo M. Amaral, Fang Fang, David Chester, Raymond Aschheim, Klee Irwin

**Affiliations:** 1Institut FEMTO-ST CNRS UMR 6174, Université de Bourgogne-Franche-Comté, F-25044 Besançon, France; 2Quantum Gravity Research, Los Angeles, CA 90290, USA

**Keywords:** DNA conformations, transcription factors, telomeres, infinite groups, free groups, algebraic surfaces, aperiodicity, character varieties

## Abstract

Taking a DNA sequence, a word with letters/bases A, T, G and C, as the relation between the generators of an infinite group π, one can discriminate between two important families: (i) the cardinality structure for conjugacy classes of subgroups of π is that of a free group on one to four bases, and the DNA word, viewed as a substitution sequence, is aperiodic; (ii) the cardinality structure for conjugacy classes of subgroups of π is not that of a free group, the sequence is generally not aperiodic and topological properties of π have to be determined differently. The two cases rely on DNA conformations such as A-DNA, B-DNA, Z-DNA, G-quadruplexes, etc. We found a few salient results: Z-DNA, when involved in transcription, replication and regulation in a healthy situation, implies (i). The sequence of telomeric repeats comprising three distinct bases most of the time satisfies (i). For two-base sequences in the free case (i) or non-free case (ii), the topology of π may be found in terms of the SL(2,C) character variety of π and the attached algebraic surfaces. The linking of two unknotted curves—the Hopf link—may occur in the topology of π in cases of biological importance, in telomeres, G-quadruplexes, hairpins and junctions, a feature that we already found in the context of models of topological quantum computing. For three- and four-base sequences, other knotting configurations are noticed and a building block of the topology is the four-punctured sphere. Our methods have the potential to discriminate between potential diseases associated to the sequences.

## 1. Introduction

Group theory and algebraic geometry serve the decipherment of ‘the book of life’ [1], a book made of a language employing four letters/nucleotides: A (adenine), T (thymine), G (guanine) and C (cytosine), as described in this work. There are finite groups, groups made of a finite number of generators and a finite number of elements that may be used to map the codons to amino acids, as carried out in our papers [2,3]. Such an approach toward the genetic code is made possible by identifying the irreducible characters of the group to the amino acids. The multiplets of codons attached to a selected amino acid correspond to the irreducible characters having the corresponding dimension of the representation (Table 3 in [2], Table 4 in [3]). A virtue of the approach is that the used irreducible characters are also seen as quantum states carrying complete quantum information.

For modeling DNA in its various conformations taken in transcription factors, telomeres and other building blocks of molecular biology, we need infinite groups defined from a motif. A sequence of the DNA nucleotides serves as the generator of the group [4]. In this context, it has been found that a group that is not free is often the witness of a potential disease. We coined the term ‘syntactical freedom’ for recognizing this property, with inspiration from an earlier work [5]. We also showed that such free groups have the distinctive property of generating an aperiodic substitution rule, providing a connection between (group) syntactical freedom and irrational numbers (Section 4 in [4]). For an infinite group, the representation cannot be based on characters but on the so-called character variety. This topic leads to a relationship between DNA, algebraic topology and algebraic geometry. Tools already proposed for topological quantum computing [6] are also used in the context of DNA conformations.

In Section 2, we briefly account for the many types of topologies that DNA can show, in terms of double strands or more strands. Then, we recall the mathematical concepts employed in our paper with some redundancy with earlier work [4,6].

In Section 3, we explain the concept of an SL(2,C) character variety associated to an infinite group with two or three generators. The former case corresponds to DNA motifs having only two distinct nucleotides. In such a case, the variety often contains the Cayley cubic associated to the Hopf link, the non disjoint union of two circles in the three-dimensional space. In the later case, the variety contains the Fricke–Klein seventh variable polynomial that is characteristic of the topology of the three-dimensional sphere with four points removed.

In Section 4, we apply these mathematical methods to transcription factors, telomeric sequences and a specific DNA decamer sequence, where almost all of its conformations have been crystallized.

## 2. Materials and Methods

Mathematical calculations performed in this paper are on the software Magma [7] (for groups) or on Sage software [8] (for character varieties).

### 2.1. DNA Conformations

DNA is a long polymer made from a chain of the nucleotides A, T, C or G. DNA exists in many possible conformations, which include a double-stranded helix of A-DNA, B-DNA and Z-DNA, although only B-DNA and Z-DNA have been directly observed in functional organisms [9,10]. The B-DNA form is most common under the conditions found in cells, but Z-DNA is often preferred when DNA binds to a protein. A view of a double helix in the A-, B- and Z-DNA forms is given in Figure 1 Other DNA conformations also exist, such as a single-stranded hairpin used mostly in macromolecular synthesis and repair, a triple-stranded H-DNA found in peptides, a G-quadruplex structure found in telomeres and a Holliday junction.

### 2.2. Finitely Generated Groups, Free Groups and Their Conjugacy Classes, and Aperiodicity of Sequences

The free group $F_{r}$ on *r* generators (of rank *r*) consists of all distinct words that can be built from *r* letters where two words are different unless their equality follows from the group axioms. The number of conjugacy classes of $F_{r}$ of a given index *d* is known and is a good signature of the isomorphism, or the closeness, of a group π to $F_{r}$. In the following, the cardinality structure of conjugacy classes of index *d* in $F_{r}$ is called the cardinality sequence (card seq) of $F_{r}$, and we need the cases from *r* = 1 to 3 to correspond to the number of distinct bases in a DNA sequence. The card seq of $F_{r}$ is in Table 1 for the three sequences of interest in the context of DNA [11].

Next, given a finitely generated group fp with a relation (rel) given by the sequence motif, we are interested in the card seq of its conjugacy classes. Often, the DNA motif in the sequence under investigation is close to that of a free group $F_{r}$, with r+1 being the number of distinct bases involved in the motif. However, the finitely generated group $f_{p}=\langle x_{1},x_{2}|rel\left(x_{1},x_{2}\right)\rangle$, or $f_{p}=\langle x_{1},x_{2},x_{3}|rel\left(x_{1},x_{2},x_{3}\right)\rangle$ or $f_{p}=\langle x_{1},x_{2},x_{3},x_{4}|rel\left(x_{1},x_{2},x_{3},x_{4}\right)\rangle$ (where the $x_{i}$ are taken in the four bases A, T, G and C, and rel is the motif), may not be the free group $F_{1}=\langle x_{1},x_{2}|x_{1}x_{2}\rangle$, or $F_{2}=\langle x_{1},x_{2},x_{3}|x_{1}x_{2}x_{3}\rangle$ or $F_{3}=\langle x_{1},x_{2},x_{3},x_{4}|x_{1}x_{2}x_{3}x_{4}\rangle$. The closeness of $f_{p}$ to $F_{r}$ can be checked by its signature in the finite range of indices of the card seq.

#### 2.2.1. Groups $f_{p}$ Close to Free Groups and Aperiodicity of Sequences

According to reference [5], aperiodicity correlates to the syntactical freedom of ordering rules. This statement was checked in the realm of transcription factors (Section 4 in [4]). Let us introduce the concept of a general substitution rule in the context of free groups. A general substitution rule ρ on a finite alphabet **𝒜**_*r*_ on *r* letters is an endomorphism of the corresponding free group $F_{r}$ (Definition 4.1 in [13]). The endomorphism property means the two relations ρ(uv)=ρ(u)ρ(v) and $\rho \left(u^{-1}\right)=\rho^{-1}\left(u\right)$, for any $u,v\in F_{r}$.

A special role is played by the subgroup $\mathrm{Aut}(F_{r})$ of automorphisms of $F_{r}$. We introduce the map $\alpha :F_{r}\to Z_{r}$ from $F_{r}$ to the Abelian group $Z_{r}$ in order to investigate the substitution rule ρ with the tools of matrix algebra.

The map α induces a homomorphism $M:\mathrm{End}\left(F_{r}\right)\to \mathrm{Mat}\left(r,Z\right)$. Under *M*, $\mathrm{Aut}(F_{r})$ maps to the general linear group of matrices with integer entries GL(r,Z). Given ρ, there is a unique mapping M(ρ) that makes the map diagram commutative [13] (p. 68). The substitution matrix M(ρ) of ρ may be specified by its elements at row *i* and column *j* as follows:$${\left(M\left(\rho \right)\right)}_{i,j}=card\left(\rho_{a_{i}}\left(a_{j}\right)\right).$$

This approach was applied to binding motifs of transcription factors [4]. The binding motif rel in the finitely presented group $fp=\langle A,T,G,C|\mathrm{rel}(A,T,G,C)\rangle$ is split into appropriate segments so that $\mathrm{rel}={\mathrm{rel}}_{A}{\mathrm{rel}}_{T}{\mathrm{rel}}_{G}{\mathrm{rel}}_{C}$ with the substitution rules $A\to rel_{A}$, $T\to rel_{T}$, $G\to rel_{G}$, $C\to rel_{C}$.

We are interested in the sequence of finitely generated groups
$$f_{p}^{\left(l\right)}=\langle A,T,G,C|\mathrm{rel}(\mathrm{rel}(\mathrm{rel}\cdots (A,T,G,C)))\rangle \;\;\left(\mathrm{with}\;\mathrm{rel}\;\mathrm{applied}\;l\;\mathrm{time}\right)$$
whose card seq is the same at each step *l* and equal to the card seq of the free group $F_{r}$ (in the finite range of indices that it is possible to check with the computer).

Under these conditions, (group) syntactical freedom correlates to the aperiodicity of sequences.

#### 2.2.2. Aperiodicity of Substitutions

There is no definitive classification of aperiodic order, the intermediate between crystalline order and strong disorder, but in the context of substitution rules, some criteria can be found. First, we need a few definitions.

A non-negative matrix M∈Mat(d,R) is one whose entries are non-negative numbers. A positive matrix *M* (denoted M>0) has at least one positive entry. A strictly positive matrix (denoted M>>0) has all positive entries. An irreducible matrix $M={\left(M_{ij}\right)}_{1\leq i,j\leq d}$ is one for which there exists a non-negative integer *k* with ${\left(M^{k}\right)}_{ij}>0$ for each pair (i,j). A primitive matrix *M* is one such that $M^{k}$ is a strictly positive matrix for some *k*.

A Perron–Frobenius (PF for short) eigenvector *v* of an irreducible non-negative matrix is the only one whose entries are positive: v>0. The corresponding eigenvalue is called the PF eigenvalue.

We will use the following criterion (Corollary 4.3 in [13]). A primitive substitution rule ρ of substitution matrix M(ρ) with an irrational PF-eigenvalue is aperiodic.

A well-studied primitive substitution rule is the Fibonacci rule $\rho =\rho_{F}:\;a\to ab,\;b\to a$ of substitution matrix $M_{F}=\left(\begin{array}{cc} 1 & 1\\ 1 & 0 \end{array}\right)$ and PF-eigenvalue equal to the golden ratio $\lambda_{PF}=\phi =\left(\sqrt{5}+1\right)/2$ (Example 4.6 in [13]). As expected, the irrationality of λ corresponds to the aperiodicity of the Fibonacci sequence.

The sequence of Fibonacci words is as follows:a,b,ab,aba,abaab,abaababa,abaababaabaab,⋯

The words have lengths equal to the Fibonacci numbers 1,1,2,3,5,8,13,21,⋯

All finitely generated groups $f_{p}^{\left(l\right)}$ whose relations rel(a,b)=ab,aba,abaab,abaababa,⋯ have a card seq whose elements are 1s, as for the card seq of the free group $F_{1}$. The Fibonacci sequence is our first example where group syntactical freedom correlates to aperiodicity.

#### 2.2.3. A Four-Letter Sequence for the Transcription Factor of the Fos Gene

Let us now apply the method to a transcription factor of importance. The transcription factor of gene Fos has selected motif rel=TGAGTCA [14]. For this case, the four-letter generated group has a card seq similar to the free group $F_{3}$ given in Table 1.

We split rel into four segments so that $\mathrm{rel}={\mathrm{rel}}_{A}{\mathrm{rel}}_{T}{\mathrm{rel}}_{G}{\mathrm{rel}}_{C}$ with the substitution maps $A\to {\mathrm{rel}}_{A}=T$, $T\to {\mathrm{rel}}_{T}=G$, $G\to {\mathrm{rel}}_{G}=AGTC$, $C\to {\mathrm{rel}}_{C}=A$ to produce the substitution sequence
A,T,G,C,ATGC,TGAGTCA,GAGTCTAGTCGAT⋯

The substitution matrix for this sequence is $M=\left(\begin{array}{cccc} 0 & 0 & 1 & 1\\ 1 & 0 & 1 & 0\\ 0 & 1 & 1 & 0\\ 0 & 0 & 1 & 0 \end{array}\right)$. It is a primitive matrix ($M^{4}>>0$) whose eigenvalues follow from the vanishing of the polynomial $\lambda^{4}-\lambda^{3}-\lambda^{2}-\lambda -1$. There are two real eigenvalues $\lambda_{1}\approx 1.92756$ and $\lambda_{2}\approx -0.77480$, as well as two complex conjugate eigenvalues $\lambda_{3,4}\approx -0.07637\pm 0.81470i$. The PF-eigenvalue is $\lambda_{PF}=\lambda_{1}$, with an eigenvector of (positive) entries $\approx {\left(1,0.37298,0.40211,0.20861\right)}^{T}$. It follows that the selected sequence for the Fos gene is aperiodic.

All of the finitely generated groups $f_{p}^{\left(l\right)}$ whose relations are
rel(A,C,G,T)=ATGC,TGAGTCA,GAGTCTAGTCGAT,⋯,
have a card seq whose elements are
1,7,41,604,13753,504243,⋯,
which is the card seq of the free group $F_{3}$. For the Fos transcription factor, group syntactical freedom correlates to aperiodicity as expected.

Further examples are obtained in the context of DNA sequences for transcription factors (Section 4 in [4]) and below, in relation to DNA conformations and telomeres.

## 3. Discussion

In the following, we make use of SL(2,C) representations of the infinite groups π arising from specific DNA sequences. The character variety has many interpretations in mathematics and physics. For instance, in mathematics, the variety is the space of representations of hyperbolic structures of three-manifolds *M* with fundamental group π(M), and the variety of the characters of SL(2,C) representations of π(M) is reflected in the algebraic geometry of the character variety [15,16]. In physics, the group SL(2,C) expresses the symmetries of fundamental physical laws. It is also known as the Lorentz group; more precisely, the double cover of the restricted Lorenz group is SL(2,C), which is the spin group.

### 3.1. SL(2,C) Character Varieties and Algebraic Surfaces

Recently, we found that the representation theory of finite groups with their character table allows us to derive an approach of the genetic code [3].

For infinite groups π such as those defined by DNA sequences, it is useful to describe the representations of π in the Lorentz group SL(2,C), the group of (2×2) matrices with complex entries and determinant 1. Such a group expresses the fundamental symmetry of all known physical laws, apart from gravitation.

Representations of π in SL(2,C) are homomorphisms ρ:π→SL(2,C) with character $\kappa_{\rho }\left(g\right)=\mathrm{tr}\left(\rho \left(g\right)\right)$, g∈π. The set of characters allows us to define an algebraic set by taking the quotient of the set of representations ρ by the group $SL_{2}\left(C\right)$, which acts by conjugation on representations [15,17].

For two-generator groups, the character variety may be decomposed into the product of surfaces, which reveals the topology of *M*. We recently found a connection between some groups whose topology is based on the Hopf link and a model of topological quantum computing [6]. The Hopf link underlies many DNA sequences whose group structure is (or is not) that of the free group $F_{1}$. The classification of the involved algebraic surfaces in the variety is performed using specific tools available in Magma [7]; see (Section 2.1 in [6]) for details.

For three-generator groups, we find that the Fricke–Klein quartic is part of the character variety.

### 3.2. The Hopf Link

Taking the linking of two unknotted curves as in Figure 2 (Left), the obtained link is called the Hopf link *H* = L2a1, whose knot group is defined as the fundamental group of the knot complement in the three-sphere $S^{3}$
(1)$$\Pi_{1}\left(S^{3}\backslash L2a1\right)=\langle a,b|[a,b]\rangle =Z^{2},$$
where [a,b]=abAB (with $A=a^{-1},\;B=b^{-1}$) is the group theoretical commutator.

There are interesting properties of the knot group $\Pi_{1}$ of the Hopf link.

First, the number of coverings of degree *d* of $\Pi_{1}$ (which is also the number of conjugacy classes of index *d*) is precisely the sum of divisor function σ(d) [20].

Second, an invariance of $\Pi_{1}$ under a repetitive action of the golden ratio substitution (the Fibonacci map) ρ:a→ab, b→a or under the silver ratio substitution ρ:a→aba, b→a exists. The terms golden and silver refer to the Perron–Frobenius eigenvalue of the substitution matrix (Examples 4.5 and 4.6 in [13]). Such an observation links the Hopf link, the group $\Pi_{1}$ of the 2-torus and aperiodic substitutions.

Using Sage software [8] developed from Ref. [17], the $SL_{2}\left(C\right)$ character variety is the polynomial corresponding to the so-called Cayley cubic
(2)$$f_{H}\left(x,y,z\right)=xyz-x^{2}-y^{2}-z^{2}+4.$$

As expected, the three-dimensional surface $\Sigma :f_{H}\left(x,y,z\right)=0$ is the trace of the commutator and is known to correspond to the reducible representations (Theorem 3.4.1 in [21]). A picture is given in Figure 3 (left).

In the perspective of algebraic geometry, we classify the homogenization of equation $f_{H}$ as a rational surface of degree 3 del Pezzo type. It displays four simple singularities.

### 3.3. Beyond the Hopf Link

As shown in [6], the Hopf link is the irreducible component of many character varieties relevant to a model of topological quantum computing. In the context of DNA groups investigated in the next section, we also find another surface with similar simple singularities as shown in Figure 3 (right). The defining polynomial is
(3)$$f_{\tilde{H}}\left(x,y,z\right)=z^{4}-2xyz\;\left(+z^{3}\right)+2x^{2}+2y^{2}-3z^{2}\left(-4z\right)-4.$$

The homogenization of equation $f_{\tilde{H}}\left(x,y,z\right)$ allows us to classify it as a conic bundle in the family of $K_{3}$ surfaces.

For the DNA sequence, whose group $\pi_{\tilde{H}}$ contains the component $f_{\tilde{H}}\left(x,y,z\right)$, we refer to the third subsection of the Results section below and the first table in this subsection. The relevant triplet nnn=CGG of the dodecameric sequence $d(CCnnnN_{6}N_{7}N_{8}GG)$ leads to a DNA conformation with the label 1ZEY in the PDB bank.

The DNA dodecamer sequence d(CCCCCGCGGGGG) is also found in the PDB bank with label 2D47, corresponding to a complete turn of A-DNA. The character variety for the group defined by this sequence contains the polynomial $f_{H}$ and a polynomial similar to $f_{\tilde{H}}$ without the third-order term $z^{3}$ and the first-order term −4z, but in the same family.

### 3.4. The Fricke–Klein Seventh Variable Polynomial

The Cayley cubic is a subset of the character variety for the four-punctured three-dimensional sphere $S_{4}^{2}$ (the sphere minus four points). Its fundamental group $\Pi_{1}$ is isomorphic to the free group $F_{3}$ of rank 3, $\Pi_{1}\left(S_{4}^{2}\right)=\langle \alpha ,\beta ,\gamma ,\delta |\alpha \beta \gamma \delta \rangle$, where the four homotopy classes α,β,γ,δ correspond to loops around the punctures.

The SL(2,C) character variety for $\Pi_{1}\left(S_{4}^{2}\right)$ satisfies a quartic equation in terms of the Fricke–Klein seventh variable polynomial [21] (p. 65) and [22]:(4)$$f\left(x,\theta \right)=xyz+x^{2}+y^{2}+z^{2}-\theta_{1}x-\theta_{2}x-\theta_{3}z+\theta_{4},$$
with $\theta_{1}=uv+wk$, $\theta_{2}=uw+vk$, $\theta_{3}=uk+vw$, $\theta_{k}=uvwk+u^{2}+v^{2}+w^{2}-4$.

## 4. Results

In this section, we apply the SL(2,C) representation theory to specific non-canonical DNA sequences having regulatory functions in gene expression (the transcription factors), replication (the telomeres) and DNA conformations.

### 4.1. Group Structure and Topology of Transcription Factors

In a transcription factor, a motif-specific DNA binding factor controls the rate of the transcription of a gene from DNA to messenger RNA by binding a protein to the DNA motif. In reference [4], we found a correlation between motifs whose subgroup structure is that of a free group and the lack of a potential disease while the gene is activated in transcription, the property of ‘syntactical freedom’.

In Table 2, this idea is illustrated by restricting to a few transcription factors whose motif comprises two bases. The card seq of the motif is either the free group $F_{1}$, close to $F_{1}$ or away from a free group when the card seq is that of the modular group $H_{3}$, of the Baumslag–Solitar group BS(−1,1) or that of groups $\pi_{1}$ and $\pi_{1}^{\prime }$. Compared to the results provided in [4], there is the additional fourth column that signals when the Groebner base for the ideal ring of the SL(2,C) character variety contains the Cayley cubic, the unique component in the case of the Hopf link [6], a degree 3 del Pezzo surface (denoted HL), or not. An additional fifth column is filled to check the presence of a surface of type $K_{3}$. Only the last row of the table for the transcription factor of gene EHF does not show this property.

#### The Character Variety for the Transcription Factor of the DBX Gene

We explicitly show the SL(2,C) character variety for the transcription factor of the DBX gene.
(5)$$\begin{array}{cc} & f_{DBX}\left(x,y,z\right)=f_{H}\left(x,y,z\right)\left(yz^{2}-y^{2}-xz-y+2\right)\left(xy^{2}-z^{3}-yz-x+3z\right)\\ & \left(y^{3}-z^{2}-3y+2\right)\left(y^{2}z-xy-yz+x-z\right)\left(z^{4}-x^{2}y+xz-4z^{2}+y+2\right) \end{array}$$

The factors in (Equation 5) are three degree 3 del Pezzo surfaces (including the Cayley cubic $f_{H}$), two rational ruled surfaces and a $K_{3}$ surface birationally equivalent to the projective plane, respectively. The latter factor also belongs to the character variety of group $\Pi_{1}\left(S^{4}\backslash {\tilde{E}}_{6}\right)$, where $S_{4}$ is the four-sphere and ${\tilde{E}}_{6}$ is the singular fiber $IV^{*}$ in Kodaira’s classification of minimal elliptic surfaces (Figure 4b in [6]).

It is important to mention that, for three-letter transcription factors, the ideal ring of the corresponding SL(2,C) character variety contains the Fricke–Klein seventh variable polynomial (Equation 4), which is a feature of the four-punctured sphere topology.

Table 3 provides a short account of the function or potential dysfunction of the genes under consideration. As mentioned before, most of the time, such a dysfunction is correlated to a card seq away from that of the free group $F_{1}$.

In view of our results, it is interesting to correlate the presence of the Hopf link HL in the character variety with the possible remodeling of B-DNA into Z-DNA or another DNA conformation. To our knowledge, general information about this subject is still lacking. From a biological point of view, it is known that some of the Z-DNA-forming conditions that are relevant in vivo are the presence of DNA supercoiling, Z-DNA-binding proteins [27] and base modifications. When transcription occurs, the movement of RNA polymerase II along the DNA strand generates positive supercoiling in front of, and negative supercoiling behind, the polymerase [28].

Perhaps the lack of HL in the character variety for transcription factors of genes in Table 2 means that the Z-DNA-forming condition is not realized.

### 4.2. Group Structure and Topology of DNA Telomeric Sequences

Terminal structures of chromosomes are made of short highly repetitive G-rich sequences with proteins known as telomeres. They have a protective role against the shortening of chromosomes through successive divisions. Most organisms use a telomere-specific DNA polymerase called telomerase that extends the 3’ end of the G-rich strand of the telomere [29]. Telomere shortening is associated with aging, mortality and aging-related diseases such as cancer.

A list of results obtained by using our group theoretical approach is in Table 4. For two-letter telomere sequences, the SL(2,C) character variety contains the Cayley cubic, the characteristic of the Hopf link HL, only in the first row. In addition to the Cayley cubic, one finds surfaces of a general type. In the next two rows, the Cayley cubic is not found. There are degree 3 del Pezzo surfaces in the factors of the character variety but not general surfaces.

As for the Hopf link, the sequence is found to be aperiodic with the Perron–Frobenius eigenvalue $\lambda_{PF}$ equal to the golden ratio. For three-letter telomere sequences, the card seq is that of the free group of $F_{2}$, except for the last row, where the identified group is $\pi_{2}$; see Figure 2 (right) for the definition of such a group. In the former seven cases, the DNA topology is known to be a G-quadruplex structure [30,31,32,33,34,35,36]. We could identify an aperiodic structure of the telomere sequence with the Perron–Frobenius eigenvalue $\lambda_{PF}$ as shown in column 5. In the latter case, the topology is of the basket type [36] and no aperiodicity of the telomere sequence could be found.

Figure 4, taken from the protein data bank (PDB 2HY9), illustrates the G-quadruplex structure of the telomere sequence in vertebrates.

**Table 4 ijms-23-13290-t004:** Group analysis of the telomere sequence found in some eukaryotes. The first column is for the telomere repeat, the second column is the organism under investigation, the third column is for the PDB code, the fourth column is for the card seq of the group π or that of the corresponding group that is identified, the fifth column is for the Perron–Frobenius eigenvalue when the sequence is found to be aperiodic, the sixth column identifies the presence of the Hopf link (in two-base sequences) or the DNA conformation (in three-base sequences) and the seventh column is a relevant reference. The notation G-quadr is for the G-quadruplex; see Figure 4. The card seq for $\pi_{1}^{\prime \prime }$ is [1,3,2,16,16,69,118,719,1877,8949⋯]. The Hecke group $H_{4}$ is defined in (Table 2 in [4]).

Seq	Organism	PDB	Card Seq	$\lambda_{\mathrm{PF}}$	Link/DNA Conf	Ref
G4T4G4	Oxytricha	1D59	$\pi_{1}^{\prime \prime }$	$(\sqrt{5}+1)/2$	HL	[37]
TG4T	universal	244D_1	$H_{4}$	.	no	[38]
T2G4	Tetrahymena	230D	$H_{4}$	.	no	[29]
T2AG3	Vertebrates	2HY9	$F_{2}$	2.5468	G-quadr.	[30]
TAG3	Giardia	2KOW	$F_{2}$	2.2055	G-quadr	[31]
T2AG2	Bombys mori	unknown	$F_{2}$	no	G-quadr	[32]
T4AG3	Green algae	unknown	$F_{2}$	3.07959	unknown	[33]
G2T2AG	Human	unknown	$F_{2}$	2.5468	G-quadr	[34]
TAG3T2AG3	Human	2HRI	$F_{2}$	3.3923	G-quadr	[35]
G3T2AG3T2AG3T	Human	unknown	$F_{2}$	4.3186	G-quadr	[36]
(GGGTTA)3G3T	Human	unknown	$\pi_{2}$	no	basket	[36]

### 4.3. Group Structure and Topology of the DNA Decamer Sequence $d(CCnnnN_{6}N_{7}N_{8}GG)$ [10]

A challenging question of structural biology is to determine if and how a DNA (or RNA) sequence defines the three-dimensional conformation, as well as the secondary and tertiary structure of proteins. In the previous two subsections, we tackled the problem with regard to transcription factors and telomeric sequences, respectively. In the former case, we restricted to the DNA part of the transcription since the DNA motif is almost exactly known from X-ray techniques while the secondary structure of the binding protein strongly depends on the model employed and the choice made to recognize the sections of the secondary structures (e.g., alpha helices, beta sheets and coils) [39]. In the latter case, in many organisms, nature invented telomerase for taking care of the replication without damaging the sequences at the 3-ends too much, while keeping the catalyzing action of DNA polymerase. Again, there is a loop complex in telomerase comprising telomere-binding proteins, with secondary structures not being analyzed so far with our group theoretical approach.

In this section, we also study DNA conformations and their relationship to algebraic topology in a specific DNA decamer sequence investigated in reference [10] by a standard crystallization technique followed by X-ray diffraction discrimination. In the sequence $d(CCnnnN_{6}N_{7}N_{8}GG)$, the factors $N_{6}$, $N_{7}$ and $N_{8}$ are taken in the two nucleotides G and C, and nnn is specified in order to maintain the self-complementarity of the sequence. This inverse repeated motif is the minimum motif used to distinguish between the double-strand forms of *B*- and *A*-DNA, while excluding the *Z*-DNA forms. A third conformation is allowed and called the four-stranded Holliday junction *J*. We refer to (Table 1 in [10]) for the main results.

On our side, the card seq of each sequence was determined and the SL(2,C) character variety was obtained. Our results are summarized in the four Table 5, Table 6, Table 7 and Table 8.

In Table 5, $N_{6}$, $N_{7}$ and $N_{8}$ are taken in the two nucleotides *G* and *C*, forming eight triplets and the associated two-letter decamer sequences. Note that the triplet CCC produces two distinct DNA conformations *A* and *J*. The character variety of the Hopf link HL (the Cayley cubic) is present in the factors of the ideal ring of the SL(2,C) character variety in five cases over the nine possibilities, where one case (with triplet CGG and code 1ZEY in the protein data bank) shows an algebraic surface similar to the Cayley cubic (with four simple singularities) as defined in Equation (Equation 3) and as shown in Figure 3 (right). We do not observe a clear correlation between the type of DNA conformation and the underlying HL topology, but the presence of HL in the variety seems to exclude the B-DNA conformation. In addition, the character variety always contains a surface of type $K_{3}$ in its factors.

In Table 6, $N_{6}$, $N_{7}$ and $N_{8}$ are taken in the two nucleotides *A* and *T*, forming eight triplets. The DNA conformation (when known) is of type *B*. The card seq that we obtain is groups $\pi_{3}^{\prime }$, $\pi_{3}^{\prime \prime }$ or $\pi_{3}^{\left(4\right)}$ as described in the caption of Table 5. In six cases over the eight possibilities, the groups encapsulate the topology of the rank 2 group $\pi_{2}$, whose associated link is shown in Figure 2 (right). As already mentioned, the SL(2,C) character variety contains the Fricke–Klein seventh variable polynomial Equation 4.

In Table 7, $N_{6}$, $N_{7}$ and $N_{8}$ are taken in the two nucleotides *A*, *G* (left part) and *A*, *C* (right part). This time, either the card seq of the group π is that of the free group $F_{3}$, of rank 3 (10 cases over the 16 possibilities), or not. In the latter case, the group encapsulates the topology of $\pi_{2}$ only at the right side of the table. Similar conclusions hold in Table 8 when $N_{6}$, $N_{7}$ and $N_{8}$ are taken in the two nucleotides *A*, *G*, *C* (left part) and *A*, *T*, *C* (right part).

To summarize this section, no clear correlation is observed between the DNA conformations of the considered decamer and our group analysis. Longer sequences may be needed to obtain such a correlation. For instance, the two-letter DNA dodecamer sequence d(CCCCCGCGGGGG) (PDB 2D47) corresponding to a complete turn of A-DNA—see Figure 5 (right)—features the polynomial $f_{H}$ (for HL) and a fourth-order polynomial similar to $f_{\tilde{H}}$ with four simple singularities, as announced at the end of Section 3.

## 5. Conclusions

In the present paper, following earlier work about the genetic code [2,3] and about the role of transcription factors in genetics [4], we made use of group theory applied to appropriate DNA motifs and we computed the corresponding variety of SL(2,C) representations. The DNA motifs under consideration may be canonical structures, such as (double-stranded) B-DNA, or non-canonical DNA structures [40], such as (single-stranded) telomeres, (double-stranded) A-DNA, Z-DNA or cruciforms, (triple-stranded) H-DNA, (four-stranded) i-motifs or G-quadruplexes, etc. One objective of the approach is to establish a correspondence between the algebraic geometry and topology of the SL(2,C) character variety and the types of canonical or non-canonical DNA-forms. For example, for two-letter transcription factors, one can correlate the presence of the Cayley cubic and/or a $K_{3}$ surface in the variety with DNA supercoiling in the remodeling of B-DNA to Z-DNA. For three- or four-letter DNA structures, our work needs to be developed in order to put the features of the variety and potential diseases in correspondence. In a separate work devoted to topological quantum computing, the topology of the four-punctured sphere and the related Fricke surfaces (generalizing the Cayley cubic) are relevant [41]. In addition, Fricke surfaces may be put in parallel with differential equations of the Painlevé VI type. It will be important to compare these models with other qualitative models based on non-linear differential equations [42,43]. In the near future, we intend to apply these mathematical tools to the context of DNA structures.

## Figures and Tables

**Figure 1 ijms-23-13290-f001:**
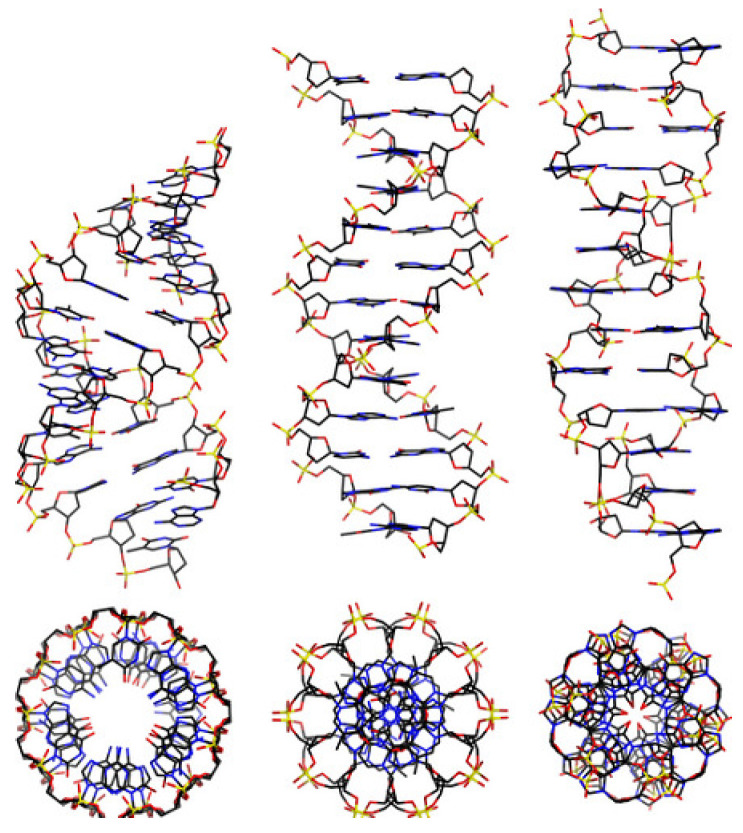
From left to right, the structures of A-, B- and Z-DNA. The view of the double helix from above (or below) shows distinct symmetries, 11-fold for the A-DNA, 10-fold for the B-DNA and 6-fold for the Z-DNA [9,10].

**Figure 2 ijms-23-13290-f002:**
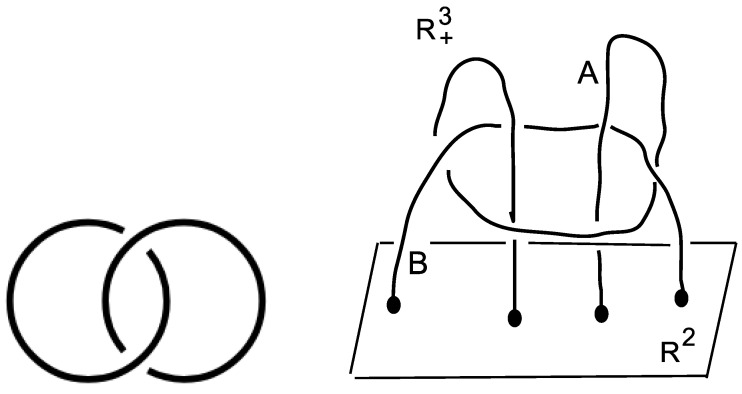
(**Left**): the Hopf link. (**Right**): the link L=A∪B is attached to the plane $R^{2}$ in the half-space $R_{+}^{3}$. It is not splittable. This can be proved by checking that the fundamental group $\pi =\pi_{2}\left(L\right)$ is not free [18] and p. 90 in [19]. One gets $\pi_{2}=\langle x,y,z|(x,(y,z))=z\rangle$, where (.,.) means the group theoretical commutator. The cardinality sequence of cc of subgroups of $\pi_{2}$ is [1,3,10,51,164,1365,9422,81594,721305,⋯] (Figure 3 in [4]).

**Figure 3 ijms-23-13290-f003:**
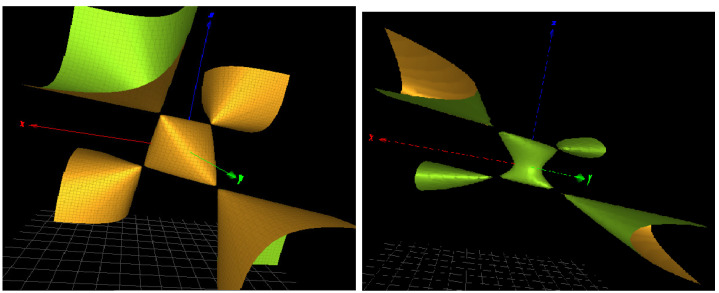
(**Left**): a three-dimensional picture of the $SL_{2}\left(C\right)$ character variety $\Sigma_{H}$ for the Hopf link complement *H*. (**Right**): a modified character variety of defining equation $f_{\tilde{H}}\left(x,y,z\right)$ with similar singularities.

**Figure 4 ijms-23-13290-f004:**
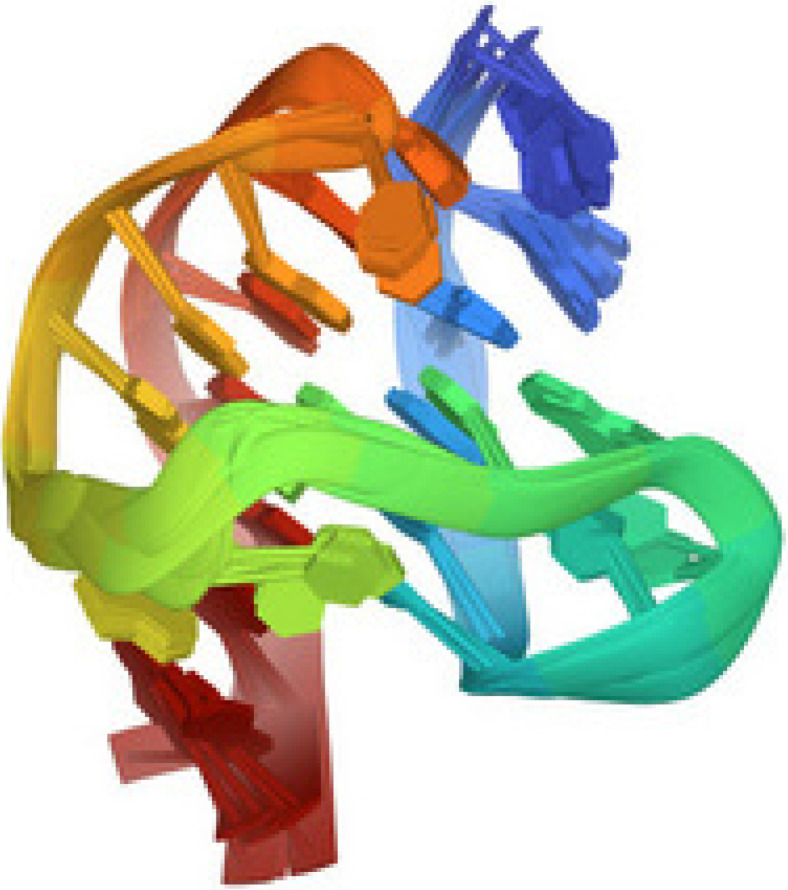
Human telomere DNA quadruplex structure in K+ solution hybrid-1 form, PDB 2HY9 [30].

**Figure 5 ijms-23-13290-f005:**
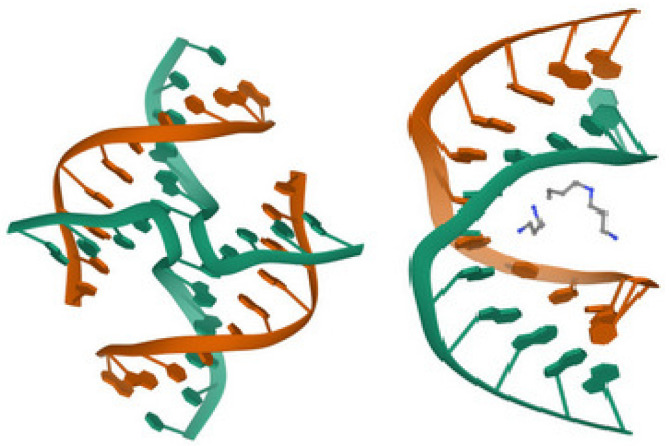
(**Left**) The four-strand Holliday junction *J*: PDB 1ZF2, (**Right**) A complete turn of A-DNA: PDB 2D47. It is associated to DNA dodecamer sequence d(CCCCCGCGGGGG) with SL(2,C) containing the factor $f_{H}=xyz-x^{2}-y^{2}-z^{2}+4$ (the Cayley cubic) and the factor $f_{\tilde{H}}=z^{4}-2xyz+2x^{2}+2y^{2}-3z^{2}-4$.

**Table 1 ijms-23-13290-t001:** Number of conjugacy classes of subgroups of index *d* in free group of rank *r* = 1 to 3 [11]. The last column is the index of the sequence in the on-line encyclopedia of integer sequences [12].

r	Card Seq	Sequence Code
1	[1,1,1,1,1,1,1,1,1,⋯]	A000012
2	[1,3,7,26,97,624,4163,34470,314493,⋯]	A057005
3	[1,7,41,604,13753,504243,24824785,1598346352,⋯]	A057006

**Table 2 ijms-23-13290-t002:** Group structure of motifs for a few two-letter transcription factors. The card seq for the modular group $H_{3}$ is [1,1,2,3,2,8,7,10,18,28,⋯]. The Baumslag–Solitar group BS(−1,1) is the fundamental group of the Klein bottle. The card seq for BS(−1,1) is [1,3,2,5,2,7,2,8,3,8,2,13,2,9,4,⋯]. The card seq for $\pi_{1}$ is [1,4,1,2,4,2,1,7,2,2,4,2,2,8,1,2,7,2,3,⋯]; for $\pi_{1}^{\prime }$, it is [1,1,1,2,1,3,3,1,2,2,1,1,9,2,14,2,1,⋯]. The symbol HL means that the Cayley cubic is part of the Groebner base for the ideal ring of the corresponding SL(2,C) character variety. For three-letter transcription factors, the ideal ring of the corresponding SL(2,C) character variety contains the Fricke–Klein seventh variable polynomial Equation 4, which is a feature of the four-punctured sphere topology. The group structure of three-letter transcription factors not leading to free groups is shown in (Table 5 in [4]).

Gene	Motif	Card Seq	Link	Type	Literature
DBX	TTTATTA	$F_{1}$	HL	$K_{3}$	[23], MA0174.1
SPT15	TATATATAT	.	.	.	., MA0386.1
PHOX2A	TAATTTAATTA	≈$F_{1}$	.	.	., MA0713.1
FOXA	TGTTTGTTT	$F_{1}$	.	.	[24,25]
FOXG	TTTGTTTTT	.	.	.	[24]
NKX6-2	TAATTAA	$H_{3}$	no	$K_{3}$	[23], [MA0675.1, MA0675.2]
FOXG	TGTTTG	BS(−1,1)	no	$K_{3}$	[23,26], MA1865.1
HoxA1, HoxA2	TAATTA	$\pi_{1}$	no	$K_{3}$	[23], [MA1495.1, MA0900.1]
POU6F1, Vax					., [MAO628.1, MA0722.1]
RUNX1	TGTGGT	.	no	.	., MA0511.1
RUNX1	TGTGGTT	$\pi_{1}^{\prime }$	no	$K_{3}$	[23], MA0002.2
EHF	CCTTCCTC	.	HL		., MA0598.1

**Table 3 ijms-23-13290-t003:** A short account of the function or dysfunction (through mutations or isoforms) of genes associated with transcription factors and sections in Table 2.

Gene	Type	Function	Dysfunction
DBX		drosophila segmentation	
SPT15	TATA-box	gene expression, regulation	
	binding protein	in Saccharomyces cerevisiae	
PHOX2A	homeodomain	differentiation, maintenance	fibrosis
		of noradrenergic phenotype	of extraocular muscles
FOX proteins	forkhead box	growth, differentiation,	
FOXA2	.	insulin secretion	diabete
		longevity	
NKX6-2	homeobox	central nervous system, pancreas	spastic ataxia
FOXG	forkhead box	notochord (neural tube)	chordoma
HoxA1	homeobox	embryonic devt of face and hear	autism
HoxA2	.	.	cleft palate
Pou6F1	.	neuroendocrine system	clear cell adenocarcinoma
Vax	.	forebrain development	craniofacial malform.
RunX1	Runt-related	cell differentiation, pain neurons	myeloid leukemia
EHF	homeobox	epithelial expression	carcinogenesis, asthma

**Table 5 ijms-23-13290-t005:** Group analysis of the sequence $d(CCnnnN_{6}N_{7}N_{8}GG)$, where $N_{6}$, $N_{7}$ and $N_{8}$ are taken in the two nucleotides *G* and *C* and nnn is specified in order to maintain the self-complementarity of the sequence [10]. The first column is for the selected triplet $N_{6}N_{7}N_{8}$, the second column is for the code in the protein data bank, the third column is for the DNA conformation when known (see Table 1 in [10]), the fourth column is for the cardinality structure of subgroups of π and the fifth column checks the occurrence of a surface corresponding to the Hopf link in the factorization of the SL(2,C) of π. The symbols A, B and J are for A-DNA, B-DNA and a four-stranded Holliday junction; lowercase is used when the conformation is not confirmed in [10].

Triplet	PDB	Conformation	Card Seq (π)	Knot
CCC	1ZF1	A	[1,1,1,1,7,1,1,2,9,6,⋯]	HL
CCC	1ZF2	J	idem	HL
CCG	1ZEX	A	idem	HL
CGG	1ZEY	A	[1,1,1,2,6,3,1,4,2,6,⋯]	HL like
CGC	none	unknown	[1,1,2,1,6,3,2,1,3,6,⋯]	no
GGG	1ZF9	A	[1,1,1,1,10,25,25,9,2,1798,⋯]	no
GCC	none	b/J	[1,1,1,1,6,1,2,1,1,6,⋯]	HL
GCG	none	unknown	[1,1,2,2,7,5,1,4,5,9,⋯]	no
GGC	none	B/a	[1,1,1,1,6,11,9,5,2,208,⋯]	no
			(card seq of Hecke group $H_{5}$)	

**Table 6 ijms-23-13290-t006:** Group analysis of the sequence $d(CCnnnN_{6}N_{7}N_{8}GG)$, where $N_{6}$, $N_{7}$ and $N_{8}$ are taken in the two nucleotides A,T [10]. Groups $\pi_{3}$ and $\pi_{3}^{\prime }$ are as in (Table 5 in [4]). The card seq for $\pi_{3}^{\prime }$ is [1,7,50,867,15906,570528,⋯]; for $\pi_{3}^{\prime \prime }$, it is [1,7,50,739,15234,548439,⋯]; for $\pi_{3}^{\left(4\right)}$, it is [1,7,59,1258,24787,⋯]. Groups $\pi_{3}^{\prime \prime }$ and $\pi_{3}^{\prime }$ may be simplified to a group whose card seq is that of $\pi_{2}$, the fundamental group of the link L=A∪B described in Figure 3 (right).

Triplet	PDB	Conformation	π
TTA	1ZFH	B	$\pi_{3}^{\prime \prime }\to \pi_{2}$
TAA	none	B	$\pi_{3}^{\prime \prime }\to \pi_{2}$
AAT	none	b	$\pi_{3}^{\prime \prime }\to \pi_{2}$
ATT	none	unknown	$\pi_{3}^{\prime \prime }\to \pi_{2}$
AAA	none	b	$\pi_{3}^{\prime }\to \pi_{2}$
TTT	none	unknown	$\pi_{3}^{\prime }\to \pi_{2}$
ATA	none	unknown	$\pi_{3}^{\left(4\right)}$
TAT	none	unknown	$\pi_{3}^{\left(4\right)}$

**Table 7 ijms-23-13290-t007:** Group analysis of the sequence $d(CCnnnN_{6}N_{7}N_{8}GG)$[10], where $N_{6}$, $N_{7}$ and $N_{8}$ are taken in the two nucleotides A,G (left) and A,C (right). Groups $\pi_{3}$ and $\pi_{3}^{\prime }$ are as in (Table 5 in [4]). The card seq for $\pi_{3}^{\left(3\right)}$ is [1,7,41,668,14969,⋯] and, for $\pi_{3}^{\left(5\right)}$, it is [1,7,41,604,28153,⋯].

Triplet	PDB	Conformation	π	Triplet	PDB	Conformation	π
AGA	1ZEW	B	$F_{3}$	ACA	none	unknown	$\pi_{3}^{\left(3\right)}\to \pi_{2}$
AGG	none	unknown	$\pi_{3}^{\left(5\right)}$	ACC	none	J	$F_{3}$
GGA	1ZFA	A	$F_{3}$	CCA	none	unknown	$F_{3}$
AAG	none	unknown	$F_{3}$	AAC	1ZF0	B	$\pi_{3}^{\prime \prime }\to \pi_{2}$
TGT	none	unknown	$F_{3}$	TCT	none	b	$\pi_{3}^{\left(3\right)}\to \pi_{2}$
TGG	1ZF6	A	$F_{3}$	TCC	none	unknown	$F_{3}$
GGT	1ZF8	A	$F_{3}$	CCT	none	b	$F_{3}$
TTG	none	unknown	$\pi_{3}^{\prime }$	TTC	none	B	$\pi_{3}^{\prime \prime }\to \pi_{2}$

**Table 8 ijms-23-13290-t008:** Group analysis of the sequence $d(CCnnnN_{6}N_{7}N_{8}GG)\;$[10], where $N_{6}$, $N_{7}$ and $N_{8}$ are taken in the three nucleotides A, G, C (left) and A, T, C (right). The card seq for $\pi_{3}^{\left(6\right)}$ is [1,7,59,874,20371,748320⋯].

Triplet	PDB	Conformation	π	Triplet	PDB	Conformation	π
AGC	1ZFM	B	$F_{3}$	ATC	1ZFC/1ZF3	B/J	$\pi_{3}^{\left(6\right)}$
ACG	none	unknown	$F_{3}$	ACT	none	B	$\pi_{3}^{\left(3\right)}\to \pi_{2}$
GCA	1ZFE	B	$F_{3}$	TCA	none	unknown	$\pi_{3}^{\left(3\right)}\to \pi_{2}$
GAC	1ZF7	B	$F_{3}$	TAC	none	unknown	$\pi_{3}^{\left(6\right)}$
CAG	none	unknown	$F_{3}$	CAT	none	unknown	$F_{3}$
CGA	none	unknown	$F_{3}$	CTA	none	unknown	$F_{3}$

## Data Availability

The datasets used and/or analyzed during the current study are available from the corresponding author on reasonable request.

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
