# Peer review of "DNA Sequence and Structure under the Prism of Group Theory and Algebraic Surfaces"

_ijms, 2022, doi:10.3390/ijms232113290_

Round 1

Reviewer 1 Report

In the paper, the authors analyzed DNA sequences and the structure of the DNA by using representations of nite groups and algebraic surfaces. A central object in the paper is the character variety of group homomorphism into SL(2, C) (up to conjugation). The results of the paper are very interesting but the representation must be improved. The following points should be considered by the authors:

  1. The paper is not easy to understand. It based on two other papers, the references [2] and [3]. The reader must read these papers to understand the motivation and the In particular, the paper [2] is central to understand the ansatz. To make the paper more self-contained, the authors should therefore include some introductory notes to explain the basics of paper [2] (the way to understand the table 3 and 4 in this paper).
  2. Beginning with section 2, the authors introduces the character variety. Here, the motivation of this approach is not well-described. In this variety, one studies the set of representations π SL(2, C) (or homomorphisms) up to conjugation, where π is the nite group. But why is it useful to consider this representation? The character variety is many interpretations in physics and At rst it is the space of at connections of a (principal) ber bundle with structure group SL(2, C) where π is the fundamental group of the underlying manifold. Secondly, it is the space of hyperbolic structures of a 3-manifold with fundamental group π (the Lorentz group is the isometry group of the hyperbolic 3-space, the authors used the double cover SL(2, C) - the spin group - which is unique by a result of Thurston). I see the intention of the authors to relate the nite group to some 3-manifold then the splitting of the 3-manifold into irreducible components (mainly hyperbolic 3-manifolds represented by knot complements) is encoded in the character variety, ref [17]). Here the authors must include more material to motivate this approach. BTW, it is not true that the Hopf link is the irreducible component of many character varieties. In [17] the splitting of π is discussed (what are the free subgroups, the Hopf link is the sum of two Z's but it doesn't mean that it enforces this splitting). This splitting is related to topological informations of the 3-manifold (splitting along incompressible surfaces). Here I recommend the paper (https://link.springer.com/article/10.1007/BF01231526) where this surface is described for knot complement in terms of the knot itself (so-called A-polynomial).
  1. I don't see how to get the DNA sequence for the polynomial (3). Even in the following sections, the authors try to demonstrate their methods for many examples. But here I don't see again a clear motivation. It seems to classify the known results with the help of nite group theory, which is OK with me but I need a clear statement of the authors why to present the table 5 to
  2. Finally, I miss a section named What was done in the paper? Was was used and what was the motivation to do it with these methods?

In principle, I recommend for the publication but only by considering the points above.

Author Response

Response to referees for the paper: DNA sequence and struc- ture under the prism of group theory and algebraic surfaces

  1. Response to Referee 1

First comment. The authors thank Referee 1 for his effort in reading our paper and for the proposals he suggests for improvement.

First, he mentions that papers 2 and 3, dealing with a new theory of the genetic code based on appropriate finite groups, could be better explained as an introduction to the present work. The representation theory for finite groups goes through their character table that encapsulates the irreducible characters. Papers 2 and 3 identify the irreducible characters with the amino acids and, in this way, the codon multiplets in the genetic code may be explained. In addition, the irreducible characters are ‘magic/fiducial states’ for informationally complete (generalized) quantum measurements (called POVMs in the quantum literature).

In contrast, in this paper, we make use of representation theory for infinite groups (instead of finite groups) and we choose representations over the (infinite) matrix group SL(2, C). The SL(2, C) representations are not used for the genetic code (the non injective map from the 64 codons to the 20 aminoacids) but for the DNA sequences involved in regularity functions of gene expression and replication (transcription factors, telomers, etc).

Since the present paper is not a review paper, we found not useful to go back to the genetic code which is a separate topic.

  • Second Second, the referee asks for a motivation of the approach based on SL(2, C) representations for the infinite groups involved in the gene expression.

We added a short introduction of Section 2 as follows and removed the corresponding words in the following subsection.

In the following, we make use of SL(2, C) representations of the infinite groups π1 arising from specific DNA sequences. The character variety has many interpretation in mathematics and physics. For instance, in mathe- matics, the variety is the space of representations of hyperbolic structures of 3-manifolds M with fundamental group π1(M ) and the variety of characters of SL(2, C) representations of π1(M ) is reflected in the algebraic geometry of the character variety [4, 5]. In physics, the group SL(2, C) expresses the symmetries of fundamental physical laws. It is also known as the Lorentz group, more precisely the double cover of the restricted Lorenz group is SL(2, C) –the spin group–.”

We are well aware of the content of paper [17] concerning of the splitting of 3-manifolds. The whole content of paper [17] is not of a straightforward utility in the context of our present work. But we followed the suggestion of referee in quoting the related paper https://link.springer.com/article/10.1007/BF01231526). By the way, we do not claim that the Hopf link is the irreducible component of many character varieties for all 3-manifolds but that we found the Hopf link in our approach of topological quantum computing in [6] and also in the context of many DNA sequences of the present paper.

Third comment. In his third comment, the referee would like a better motivation for the section3: Results.

We added a short sentence at the beginning of this section. “

In this section, we apply the SL(2, C) representation theory to specific non-canonical DNA sequences having regulatory functions in gene expression

– the transcription factors–, replication –the telomeres– and DNA confor- mations.”

We also sligthly modify the sentence about the polynomial (3) which is not fully clear.

For the DNA sequence, whose group πH˜  contains the component fH˜ (x, y, z), we refer to the third subsection of section Results below and the first table in this subsection. The relevant triplet nnn=CGG of the dodecameric se-

quence d(CCnnnN6N7N8GG) leads to a DNA conformation with the label 1ZEY in the PDB bank.

Fourth comment. The referee asks a conclusion. A conclusion was added to the work as follows.

“In the present paper, following earlier work about the genetic code [1, 2] and about the role of transcription factors in genetics [3], we make use of group theory applied to appropriate DNA motifs and we compute the corre- sponding variety of SL(2, C) representations. The DNA motifs under con- sideration may be canonical structures such as (double-stranded) B-DNA or non-canonical DNA structures [7] such as (single-stranded) telomeres, (double-stranded) A-DNA, Z-DNA or cruciforms, (triple-stranded) H-DNA, (four-stranded) i-motifs or G-quadruplexes, etc. One objective of the ap- proach is to establish a correspondence between the algebraic geometry and topology of the SL(2, C) character variety and the types of canonical or non- canonical DNA-forms. For example, for two-letter transcription factors, one can correlate the presence of the Cayley cubic and/or a K3 surface in the variety with DNA supercoiling in the remodeling of B-DNA to Z-DNA. For three- or four-letter DNA structures, our work needs to be developed in order to put in correspondence the features of the variety and potential diseases. In a separate work devoted to topological quantum computing, it happens that the topology of the four-punctured sphere and the related Fricke sur- faces (generalizing the Cayley cubic) are relevant [8]. In addition, Fricke surfaces may be put in parallel with differential equations of the Painlev’e VI type.  It will be important to compare these models with other quali- tative models based on non linear differential equations [9, 10]. In a near future, we intend to apply these mathematical tools to the context of DNA structures.

References

  • Planat, M.; Aschheim, R.; Amaral, M. M.; Fang, F.; Irwin, K. Complete quantum information in the DNA genetic code . Symmetry 2020 12,
  • Planat, ; Chester, D.; Aschheim, R.; Amaral, M.M.; Fang, F.; Irwin,  K.  Finite groups for the Kummer surface: The genetic code and quantum gravity. Quantum Rep. 2021 3, 68–79.
  • Planat, ; Aschheim, R.; Amaral, M. M.; Fang F.; Chester, D.;  Irwin  K.  Group theory of syntactical freedom in DNA transcription and genome decoding. Curr. Issues Mol. Biol. 2022 44, 1417–1433.
  • Culler, ; Shalen P. B. Varieties of group representations and splitting of 3- manifolds, Ann. of Math. 1983, 117, 109–146.
  • Cooper, ; Culler, M.; Gillet, H.; Long, D. D.; Shalen, P. B. Plane curves associated to character varieties of 3-manifolds. Invent. Math. 1994 118, 47–84.
  • Planat, M.; Aschheim, R.; Amaral, M. M.; Fang F.; Chester, D.; Irwin K. Character varieties and algebraic surfaces for the topology of quantum computing. Symmetry 2022 14,
  • Bansal,  ;   Kaushik,   S.;   Kukreti,   S.   Non-canonical   DNA   structures:    di- versity and disease association. Front. Genet. 2022, 05 September 2022 (https://doi.org/10.3389/fgene.2022.959258).
  • Planat, ; Chester, D.; Amaral, M.; Irwin K. Fricke topological qubits. Preprints 2022, (doi: 10.20944/preprints202210.0125.v1).
  • Matsutani, ; Previato, E. An algebro-geometric model for th shape of supercolied DNA. Physica D: Nonlinear Phenomena 2022 430 133073.
  • Rand, A., Raju, A.; Saez, M.; Siggia, E. D. Geometry of gene regulatory dynamics. PNAS 2021 13 September 2021 (https://doi.org/10.1073/pnas.2109729118).

Reviewer 2 Report

The paper applies quantum math & physics to the generally accepted biological (bioinformatical) view, that non-canonical DNA structures have regulatory functions in gene expression and replication. The authors already applied their approach to the so called genetic code, defining the transcription and translation of the four letter triplet code of DNA into the 20 letter amino acid "language".

However, describing the so-called non-canonical DNA structures by the authors' approach, is far more complicated, though most appropriate and innovative. For the first time - and I am covering decades of my own experience in the field of chromatin structure and gene expression - molecular biology could become lifted up from describing just "quality" into a new orbit, where the interplay between quality, quantity, and time in the regulation of gene expression becomes defined my mathematics. I therefor definitely want this paper being published!

Referred to the mathematical background, well established notions from algebraic geometry combined with topological aspects were applied. As far as visible, I couldn't detect any draw-backs concerning the rigor of the mathematical argumentations, though several of them are just covered by citations.

Now the "however" sentences: There may be not many physicists, mathematicians, and/or biologists who understand both sides of the coin. That's why I recommend to cite and discuss by adding the "qualitative figures" from one of the many comprehensive review papers on non-canonical DNA structures. Also, from the "conventional" mathematical side, I would like to see papers added, like:

An algebro-geometric model for the shape of supercoiled DNA (https://doi.org/10.1016/j.physd.2021.133073)

Geometry of gene regulatory dynamics (https://doi.org/10.1073/pnas.2109729119)

Author Response

  1. Response to Referee 2

We thank Referee 2 about his reading. We completely agree with him that our paper is in the direction of confirming “the generally accepted view that non-canonical DNA structures have regulatory functions in gene expression and regulation”. Our reply to the “however” sentence is a conclusion that was lacking in our first submission. In the conclusion, we quote a recent review paper by Bansal et al about non-canonical DNA structures. We also quote the two proposed papers by Matsutani et al, and Rand et al.  This work  may  have  to  do  with  the  correspondence  between  Painlev´e  VI  and Fricke surfaces, as mentioned at the end.

Round 2

Reviewer 1 Report

The author adress all my comments. The paper is now more readiable. The changes in the conclusion were absolutely necessary to understand the work in this paper.